# Order-Optimal Regret in Distributed Kernel Bandits using Uniform Sampling with Shared Randomness

**Nikola Pavlovic**[*]
Department of Electrical and Computer Engineering
Cornell University
Ithaca, NY, USA
np358@cornell.edu

**Sudeep Salgia**
Department of Electrical and Computer Engineering
Carnegie Mellon University
Pittsburgh, PA, USA
ssalgia@andrew.cmu.edu

**Qing Zhao**
Department of Electrical and Computer Engineering
Cornell University
Ithaca, NY, USA
qz16@cornell.edu

## Abstract

We consider distributed kernel bandits where $N$ agents aim to collaboratively maximize an unknown reward function that lies in a reproducing kernel Hilbert space. Each agent sequentially queries the function to obtain noisy observations at the query points. Agents can share information through a central server, with the objective of minimizing regret accumulated over time and agents. We develop the first algorithm that achieves the optimal regret order with a communication cost that is sublinear in both $N$ and $T$. The key features of the proposed algorithm are the uniform exploration at the local agents and shared randomness with the central server. Working together with the sparse approximation of the GP model, these two approaches make it possible to preserve the learning rate of the centralized setting at a diminishing rate of communication.

## 1 Introduction

**Distributed Kernel Bandits.** We study the problem of zeroth-order online stochastic optimization in a distributed setting, where $N$ agents aim to collaboratively maximize a reward function with communications facilitated by a central server. The reward function $f : \mathcal{X} \to \mathbb{R}$ is unknown, we only assume it belongs to a Reproducing Kernel Hilbert Space (RKHS) associated with a known kernel $k$. At each time instant $t$, each agent $n$ chooses a point $x_t^{(n)} \in \mathcal{X}$ and receives noisy feedback on the function value at the query point. The goal is for each distributed agent to converge to $x^* \in \arg\max_{x \in \mathcal{X}} f(x)$, a global maximizer of $f$. We quantify this goal as minimizing the

---

[*]This work was supported in part by the USDA/NSF AI Institute for Next Generation Food Systems under USDA award number 2020-67021-32855.

Workshop on Bayesian Decision-making and Uncertainty, 38th Conference on Neural Information Processing Systems (NeurIPS 2024).

cumulative regret summed over a learning horizon of length $T$ and over all $N$ agents: $R(T) = \sum_{n=1}^{N} \sum_{t=1}^{T} (f(x^*) - f(x_t^{(n)}))$.

In addition to learning efficiency, distributed kernel bandits face a new challenge of communication efficiency. Without constraints on the communication cost, all agents can share their local observations and coordinate their individual query actions at no cost. At the other end of the spectrum is a complete decoupling of the agents, resulting in $N$ independent single-user problems without the benefit of data sharing for accelerated learning. The tension between learning efficiency and communication efficiency is evident. A central question to this trade-off is how to achieve the optimal learning rate enjoyed by the centralized setting using a minimum amount of message exchange among agents.

**Main Results.** In this paper, we develop the first algorithm for distributed kernel bandits that achieves the optimal order of regret enjoyed by centralized learning with a sublinear message exchange in both $T$ and $N$.

To tackle the essential trade-off between learning rate and communication efficiency, a distributed learning algorithm needs a communication strategy that governs *what* to communicate and *how to integrate* the shared information into local policies. To minimize the total regret that is accumulating over time and agents, the communication strategy needs to work in tandem with the query actions to ensure a continual flow of information available at all agents for decision-making.

A natural answer to *what* to communicate in a distributed learning problem is a certain sufficient local statistics of the underlying unknown parameters [31]. However, for kernel bandits, relevant sufficient statistics are infinite-dimensional and hence an impractical choice for communication. Existing studies resolve this issue by exchanging local query actions and observations across all agents and throughout the learning horizon [9, 15], resulting in a communication cost growing linearly in both $N$ and $T$. Even with a communication cost growing linearly in both $N$ and $T$ it is not clear how to achieve the performance of a centralized learner with $NT$ query points. Prevailing approaches in centralized kernel bandits utilize reward-dependent [28] or adaptive policies [16]. Emulating such policies at each of the $NT$ query points is practically unfeasible as it would require the agents to *take turns* in their queries and immediately share the local observations with all other agents.

To tackle the above challenges, our proposed algorithm represents major departures from the prevailing approaches. Referred to as DUETS (Distributed Uniform Exploration of Trimmed Sets), this algorithm has two key features: *uniform exploration* at the local agents and *shared randomness* with the central server. In DUETS , each agent employs uniform sampling as the query strategy. Uniform sampling is fully compatible with parallel learning. In particular, note that the union of the local sets of size $t$ query points obtained at the agents through uniform sampling is identical (in distribution) to the set of size $Nt$ query points obtained at a centralized decision maker using the same uniform sampling strategy. This superposition property of uniform sampling allows us to leverage the recent results on random exploration in centralized kernel bandits [29], and is crucial in achieving the optimal learning rate defined by the centralized setting. In terms of communication efficiency, uniform sampling makes it possible to bypass the exchange of query points altogether and reduce the exchange of reward observations through the *shared randomness* strategy. In DUETS, each agent has access to an independent coin, i.e., a source of randomness, which is unknown to the other agents but is known to the server. The shared randomness enables the server to reproduce the points queried by the agents, thereby resulting in effective transmission of the local set of queried points to the server at *no communication cost*. Please refer to App. C for an additional discussion.

We analyze the performance of DUETS and establish that it incurs a cumulative regret of $\widetilde{\mathcal{O}}(\sqrt{NT\gamma_{NT}} \log(T/\delta))$ with probability $1 - \delta$, where $\gamma_{NT}$ denotes the maximal information gain of the kernel Srinivas et al. [28] and $\tilde{\mathcal{O}}(\cdot)$ hides poly-logarithmic factors. To the best knowledge of the authors, this is the *first* algorithm to achieve the optimal order of regret for the problem of distributed kernel bandits. We also establish a bound of $\tilde{\mathcal{O}}(\gamma_{NT})$ on the communication cost incurred by DUETS (See Section 2 for a precise definition) .

**Related Work.** The existing literature on distributed kernel bandits is relatively slim. The most relevant to our work is that by Li et al. [15], where the authors consider the problem of distributed contextual kernel bandits and propose a UCB based policy with sparse approximation of GP models and intermittent communication. Their proposed policy was shown to incur a cumulative regret

of $\widetilde{\mathcal{O}}(\sqrt{NT}\gamma_{NT})$ and communication cost of $\mathcal{O}(N\gamma_{NT}^3)$. The DUETS algorithm proposed in this work, offers an improvement over the algorithm in [15] both in terms of regret and communication cost. While the contextual setting with varying arm action sets considered in their work is more general that the setting with a fixed arm set considered in this work, their proposed algorithm does not offer non-trivial reduction in regret or communication cost in the fixed arm setting. Moreover, both the regret and communication cost incurred by the algorithm in Li et al. [15] are not guaranteed to be sublinear in the total number of queries, $NT$, for all kernels. Consequently, their algorithm does not guarantee convergence to $x^*$ or a non-trivial communication cost for all kernels. On the other hand, both regret and communication cost of DUETS is guaranteed to be sub-linear implying both convergence and communication efficiency.

Among other studies, Du et al. [8] consider the problem of distributed pure exploration in kernel bandits over finite action set, where they focus on designing learning strategies with low simple regret. In this work, we consider the more challenging continuum-armed setup with a focus on minimizing cumulative regret as opposed to simple regret. Another line of work explores impact of heterogeneity among clients and design algorithms to minimize this impact. Salgia et al. [21] consider personalized kernel bandits in which agents have heterogeneous models and aim to optimize the weighted sum of their own reward function and the average reward function over all the agents. Dubey and Pentland [9] consider heterogeneous distributed kernel bandits over a graph in which they use additional kernel-based modeling to measure task similarity across different agents.

In contrast to the distributed kernel bandit, the problems of distributed multi-armed bandits and linear bandits have been extensively studied. For distributed multi-armed bandits (MAB), a variety of algorithms have been proposed for distributed learning under different network topologies Landgren et al. [14], Shahrampour et al.[25], Sankararaman et al.[23], Chawla et al.[6], Zhu et al.[33]. Shi et al. [27] and Shi and Shen [26] have analyzed the impact of heterogeneity among agents in the distributed MAB problem. Similarly, the problem of distributed linear bandits is also well-understood in variety of settings with different network topologies Korda et al, [13], heterogeneity among agents Mitra et al. [17], Ghosh et al.[10], Hanna et al.[11] and communication constraints Mitra et al. [18], Wang et al.[31], Huang et al.[12], Amani et al.[3], Salgia and Zhao[22].

## 2 Problem Formulation

We consider a distributed learning framework consisting of $N$ agents indexed by $\{1, 2, \ldots, N\}$. Under this framework, we study the problem of collaboratively maximizing an unknown function $f : \mathcal{X} \to \mathbb{R}$, where $\mathcal{X} \subset \mathbb{R}^d$ is a compact, convex set. The function $f$ belongs to the an RKHS, $\mathcal{H}_k$, associated with a known positive definite kernel $k : \mathcal{X} \times \mathcal{X} \to \mathbb{R}$. $\mathcal{H}_k$ is a Hilbert space that is endowed by with an inner product $\langle \cdot, \cdot \rangle_{\mathcal{H}_k}$ that obeys the reproducing property, and induces the norm $\|g\|_{\mathcal{H}_k} = \langle g, g \rangle_{\mathcal{H}_k}$. We assume $f$ is finite in this norm i.e $\|f\|_{\mathcal{H}_k} \leq B$.

Each agent, upon querying a point $x \in \mathcal{X}$, observes $y = f(x) + \epsilon$, where $\epsilon$ is a zero mean, $R$-sub Gaussian noise term assumed to be independent across time and agents. We make the following assumption on the unknown function $f$, similar to that adopted in Salgia et al [29].

**Assumption 2.1.** Let $\mathcal{L}_\eta = \{x \in \mathcal{X} | f(x) \geq \eta\}$ denote the level set of $f$ for $\eta \in [-B, B]$. We assume that for all $\eta \in [-B, B]$, $\mathcal{L}_\eta$ is a disjoint union of at most $M_f < \infty$ components, each of which is closed and connected. Moreover, for each such component, there exists a bi-Lipschitzian map between each such component and $\mathcal{X}$ with normalized Lipschitz constant pair $L_f, L'_f < \infty$.

The communication efficiency is measured using the sum of the uplink and downlink communication costs. In particular, let $C_{\text{up}}^{(n)}(T)$ denote the number of real numbers sent by the agent $n$ to the server over the time horizon. The uplink cost of $\pi$ is given as $C_{\text{up}}^\pi(T) = \frac{1}{N} \sum_{n=1}^N C_{\text{up}}^{(n)}(T)$. Similarly, the downlink cost of $\pi$, $C_{\text{down}}^\pi(T)$ is the number of real numbers broadcast by the server over the entire time horizon averaged over all agents. The overall communication cost of $\pi$ is given by $C^\pi(T) = C_{\text{up}}^\pi(T) + C_{\text{down}}^\pi(T)$.

### 2.1 GP Models

A Gaussian Process (GP) is a random process $G$ indexed by $\mathcal{X}$ and is associated with a mean function $\mu : \mathcal{X} \to \mathbb{R}$ and a positive definite kernel $k : \mathcal{X} \times \mathcal{X} \to \mathbb{R}$. The random process

$G$ is defined such that for all finite subsets of $\mathcal{X}$, $\{x_1, x_2, \ldots, x_m\} \subset \mathcal{X}$, $m \in \mathbb{N}$, the random vector $[G(x_1), G(x_2), \ldots, G(x_m)]^\top$ follows a multivariate Gaussian distribution with mean vector $[\mu(x_1), \ldots, \mu(x_n)]]^\top$ and covariance matrix $\Sigma = [k(x_i, x_j)]_{i,j=1}^m$. Throughout the work, we consider GPs with $\mu \equiv 0$. When used as a prior for a data generating process under Gaussian noise, the conjugate property provides closed form expressions for the posterior mean and covariance of the GP model. Specifically, given a set of observations $\{\mathbf{X}_m, \mathbf{Y}_m\} = \{(x_i, y_i)\}_{i=1}^m$ from the underlying process, the expression for posterior mean and variance of GP model is given as follows:

$$\mu_m(x) = k_{\mathbf{X}_m}(x)^\top (\lambda \mathbf{I}_m + \mathbf{K}_{\mathbf{X}_m, \mathbf{X}_m})^{-1} \mathbf{Y}_m, \tag{1}$$

$$\sigma_m^2(x) = (k(x, x) - k_{\mathbf{X}_m}^\top(x)(\lambda \mathbf{I}_m + \mathbf{K}_{\mathbf{X}_m, \mathbf{X}_m})^{-1} k_{\mathbf{X}_m}(x)). \tag{2}$$

In the above expressions, $k_{\mathbf{X}_m}(x) = [k(x_1, x), k(x_2, x) \ldots k(x_n, x)]^\top$, $\mathbf{K}_{\mathbf{X}_m, \mathbf{X}_m} = \{k(x_i, x_j)\}_{i,j=1}^m$, $\mathbf{I}_m$ is the $m \times m$ identity matrix and $\lambda$ is the variance of the Gaussian noise.

Following a standard approach in the literature [28], we model the data corresponding to observations from the unknown $f$, which belongs to the RKHS of a positive definite kernel $k$, using a GP with the same covariance kernel $k$. [2] The benefit of this approach is that the posterior mean and variance of this GP model serve as tools to both predict the values of the function $f$ and quantify the uncertainty of the prediction at unseen points in the domain [30, Thm. 1].

**Sparse approximation of GP models.** The sparsification of GP models refers to the idea of approximating the posterior mean and variance of a GP model, corresponding to a set of observations $\{\mathbf{X}_m, \mathbf{Y}_m\}$, using a subset of query points $\mathbf{X}_m$. In particular, let $\mathcal{S}$ be a subset of $\mathbf{X}_m$ consisting of $r < m$ points. The approximate posterior mean and variance [32] based on points in $\mathcal{S}$, referred to as the inducing set, is given as

$$\tilde{\mu}_m(x) = z_{\mathcal{S}}(x)^\top \left( \lambda \mathbf{I}_{|\mathcal{S}|} + \mathbf{Z}_{\mathbf{X}_m, \mathcal{S}}^\top \mathbf{Z}_{\mathbf{X}_m, \mathcal{S}} \right)^{-1} \mathbf{Z}_{\mathcal{X}_m, \mathcal{S}}^\top \mathbf{Y}_m \tag{3}$$

$$\lambda \tilde{\sigma}_m^2(x) = \left[ k(x, x) - z_{\mathcal{S}}^\top(x) \mathbf{Z}_{\mathbf{X}_m, \mathcal{S}}^\top \mathbf{Z}_{\mathbf{X}_m, \mathcal{S}} \left( \lambda \mathbf{I}_{|\mathcal{S}|} + \mathbf{Z}_{\mathbf{X}_m, \mathcal{S}}^\top \mathbf{Z}_{\mathbf{X}_m, \mathcal{S}} \right)^{-1} z_{\mathcal{S}}(x) \right], \tag{4}$$

where $z_{\mathcal{S}}(x) = \mathbf{K}_{\mathcal{S}, \mathcal{S}}^{-\frac{1}{2}} k_{\mathcal{S}}(x)$ and $\mathbf{Z}_{\mathbf{X}_m, \mathcal{S}} = [z_{\mathcal{S}}(x_1), z_{\mathcal{S}}(x_2), \ldots, z_{\mathcal{S}}(x_m)]^\top$.

## 3 The DUETS Algorithm

We first describe the randomization at each agent and the shared randomness with the server. Each agent $n$ has a coin $\mathscr{C}_n$ for generating random bits that are independent of those generated by other agents. Each agent's coin is unknown to other agents, but known to the central server. In a practical implementation, the coins can be thought of as seeds for generating random numbers.

DUETS employs an epoch-based elimination structure where the domain $\mathcal{X}$ is successively trimmed across epochs to maintain an active region that contains a global maximizer $x^*$ with high probability. Specifically, in each epoch $j$, the server and the agents maintain a common active subset of the domain $\mathcal{X}_j \subseteq \mathcal{X}$ with $\mathcal{X}_1$ initialized to $\mathcal{X}$. The operations in each epoch are as follows.

During the $j^{\text{th}}$ epoch, each agent $n$, using its private coin $\mathscr{C}_n$, generates $\mathcal{D}_j^{(n)}$, a set of $T_j = \lfloor \sqrt{T T_{j-1}} \rfloor$ points that are uniformly distributed in the set $\mathcal{X}_j$.[3] Each agent $n$ queries all the points in $\mathcal{D}_j^{(n)}$ and obtains $\mathbf{Y}_j^{(n)} \in \mathbb{R}^{T_j}$, the corresponding vector of reward observations. Since the server has access to the coins of all the agents, it can faithfully reproduce the set $\mathcal{D}_j = \bigcup_{n=1}^N \mathcal{D}_j^{(n)}$ without any communication between the server and the agents. In order to efficiently communicate the observed reward values from the agents to the server, we leverage sparse approximation of GP models along with the knowledge of the set $\mathcal{D}_j$ at the server. The server constructs a global inducing set $\mathcal{S}_j$ by including each point in $\mathcal{D}_j$ with probability $p_j := p_0 \sigma_{j,\max}^2$, independent of other points where $\sigma_{j,\max}^2 = \sup_{x \in \mathcal{X}_j} \sigma_j^2(x)$, $\sigma_j^2(\cdot)$ is the posterior variance corresponding to points collected in $\mathcal{D}_j$ and $p_0$ is an appropriately chosen constant. The server broadcasts the inducing set $\mathcal{S}_j$ to all the agents.

---

[2] We assume a *fictitious* GP prior over the fixed, unknown function $f$ along with *fictitious* Gaussian noise.

[3] If the active region consists of multiple disjoint regions, then we carry out this step for each region separately. For simplicity of description, we assume the active region consists of a single connected component.

Upon receiving the inducing set, each agent $n$ computes $v_j^{(n)} := \mathbf{Z}_{\mathcal{D}_j^{(n)}, \mathcal{S}_j}^\top \mathbf{Y}_j^{(n)} \in \mathbb{R}^{|\mathcal{S}_j|}$, the projection of its reward vector onto the inducing set. All agents then send the projected observations $v_j^{(n)}$ to the server, which aggregates them to obtain the vector $\overline{v}_j := (\lambda \mathbf{I}_{|\mathcal{S}_j|} + \mathbf{Z}_{\mathcal{D}_j, \mathcal{S}_j}^\top \mathbf{Z}_{\mathcal{D}_j, \mathcal{S}_j})^{-1} (\sum_{n=1}^N v_j^{(n)})$. Note that the summation $\sum_{n=1}^N v_j^{(n)}$ equals to $\mathbf{Z}_{\mathcal{D}_j, \mathcal{S}_j}^\top \mathbf{Y}_j$, i.e., projection of the rewards of all agents onto the inducing set. The server then broadcasts the vector $\overline{v}_j$ and $\sigma_{j,\max}$ to all the agents. The benefit of sending $\overline{v}_j$ as opposed to the sum of rewards is that it allows the agents to compute the posterior mean at the agents using their knowledge of the inducing set $\mathcal{S}_j$ (See. Eqn (3)).

As the last step of the epoch, all the agents and the server trim the current set $\mathcal{X}_j$ to $\mathcal{X}_{j+1}$ using the update rule: $\mathcal{X}_{j+1} = \left\{ x \in \mathcal{X}_j : \tilde{\mu}_j(x) \geq \sup_{x' \in \mathcal{X}_j} \tilde{\mu}_j(x') - 2\beta(\delta')\sigma_{j,\max} \right\}$, where $\delta' = \frac{\delta}{2|\mathcal{U}_T| \cdot (\log(\log N \log T)) + 4)}$ and $\tilde{\mu}_j(x) = z_{\mathcal{S}_j}^\top(x)\overline{v}_j$ is the *approximate* posterior mean computed based on the inducing set $\mathcal{S}_j$ (See Eqn. (3)). Below we present the pseudo code for DUETS on the agent's side. For pseudo-code for the server side please refer to App. A .

---

**Algorithm 1** DUETS : Agent $n \in \{1, 2, \ldots, N\}$

---

1: **Input**: Size of the first epoch $T_1$, error probability $\delta$
2: $t \leftarrow 0, j \leftarrow 1, \mathcal{X}_1 \leftarrow \mathcal{X}$
3: **while** $t < T$ **do**
4:     $\mathcal{D}_j^{(n)} = \emptyset$
5:     **for** $i \in \{1, 2, \ldots, T_j\}$ **do**
6:         Query a point $x_t^{(n)}$ uniformly at random from $\mathcal{X}_j$ using the coin $\mathscr{C}_n$ and observe $y_t^{(n)}$
7:         $\mathcal{D}_j^{(n)} \leftarrow \mathcal{D}_j^{(n)} \cup \{x_t^{(n)}\}, t \leftarrow t + 1$
8:         **if** $t > T$ **then** Terminate
9:     **end for**
10:     Receive the global inducing set $\mathcal{S}_j$,
11:     Set $v_j^{(n)} \leftarrow \mathbf{Z}_{\mathcal{D}_j^{(n)}, \mathcal{S}_j}^\top \mathbf{Y}_j^{(n)}$, where $\mathbf{Y}_j^{(n)} = [y_{t-T_j}, y_{t-T_j+1}, \ldots, y_t]^\top$
12:     Receive $\overline{v}_j$ and $\sigma_{j,\max}$ from the server and compute $\tilde{\mu}_j(\cdot) = z_{\mathcal{S}_j}^\top(\cdot)\overline{v}_j$
13:     Update $\mathcal{X}_j$ to $\mathcal{X}_{j+1}$ using Eqn. (3) ,$T_{j+1} \leftarrow \lfloor \sqrt{T T_j} \rfloor, j \leftarrow j + 1$
14: **end while**

---

**Performance guarantees.**   The following theorem characterizes the regret performance and communication cost of DUETS.

**Theorem 3.1.** *Consider the distributed kernel bandit problem described in Section 2. For a given $\delta \in (0, 1)$, let the policy parameters of* DUETS *be such that $T_1 \geq \overline{M}/N$ and $p_0 = 72 \log \frac{4N}{\delta}$. Then with probability at least $1 - \delta$, the regret and communication cost incurred by* DUETS *satisfy the following relations:*

$$R_{\text{DUETS}} = \tilde{\mathcal{O}}(\sqrt{NT\gamma_{NT}} \log(T/\delta)); \quad C_{\text{DUETS}} = \tilde{\mathcal{O}}(\gamma_{NT}).$$

In the above theorem, $\overline{M}$ is a constant that is independent of $N$ and $T$. As shown in above theorem, DUETS achieves order-optimal regret as it matches the lower bound established in [24] upto logarithmic factors. DUETS is the *first algorithm* to close this gap to the lower bound in the distributed setup and achieve order-optimal regret performance. We refer the reader to App. B for a detailed proof of the theorem.

## 4   Conclusion

We propose a new algorithm for the problem of distributed kernel bandits. The proposed algorithm represents major departures from prevailing approaches and has two key features: uniform exploration and shared randomness. It is the *first* algorithm to achieve optimal-order regret in distributed kernel bandit setting while simultaneously achieving diminishing rates of communication in both the time horizon and the number of agents. We also corroborate our theoretical claims with empirical studies (see App. D).

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

# A Additional details about DUETS

Below we outline the pseudo-code of DUETS from the server and agent side.

---
**Algorithm 2** DUETS : Server
---
1: **input**: Size of the first epoch $T_1$, error probability $\delta$
2: $t \leftarrow 0, j \leftarrow 1, \mathcal{X}_1 \leftarrow \mathcal{X}$
3: **while** $t < T$ **do**
4:      Use the coins $\mathscr{C}_1, \mathscr{C}_2, \ldots, \mathscr{C}_N$ to reproduce the sets $\mathcal{D}_j^{(1)}, \mathcal{D}_j^{(2)}, \ldots, \mathcal{D}_j^{(N)}$
5:      $\mathcal{D}_j \leftarrow \cup_{n=1}^N \mathcal{D}_j^{(n)}$
6:      Set $\sigma_{j,\max} \leftarrow \sup_{x \in \mathcal{X}_j} \sigma_j(x)$
7:      Construct the set $\mathcal{S}_j$ by independently including each point from $\mathcal{D}_j$ with probability $p_j$
8:      Broadcast $\mathcal{S}_j$ to all the agents and receive $v_j^{(n)}$ from all agents $n \in \{1, 2, \ldots, N\}$
9:      Set $\overline{v}_j$ using Eqn. (3)
10:     Broadcast $\overline{v}_j$ and $\sigma_{j,\max}$ to all the agents
11:     Update $\mathcal{X}_j$ to $\mathcal{X}_{j+1}$ using Eqn. (3)
12:     $t \leftarrow t + T_j, T_{j+1} \leftarrow \lfloor \sqrt{TT_j} \rfloor$
13:     $j \leftarrow j + 1$
14: **end while**

---

# B Proof of Theorem 3.1

In this section, we provide a detailed proof of Theorem 3.1. For the regret bound, we first bound the regret incurred by DUETS in each epoch $j$ and then sum it across different epochs to obtain a bound on the overall cumulative regret. We first prove the theorem assuming the results from Lemmas B.1, B.2 and B.3 and then separately prove the lemmas.

**Lemma B.1.** *Let $\Delta_j := \sup_{x \in \mathcal{X}_j} f(x^*) - f(x)$. Then, the following bound holds all epochs $j \geq 1$ with probability $1 - \delta/2$.*

$$\Delta_j \leq 8\beta(\delta') \cdot \sup_{x \in \mathcal{X}_{j-1}} \sigma_j(x) + \frac{4B}{T},$$

*where $\delta' = \frac{\delta}{2(\log(\log N + \log T) + 4)|\mathcal{U}_T|}$ and $\mathcal{U}_T$ denotes the discretization defined in Assumption B.8.*

**Lemma B.2.** *[29] Let $\sigma_j^2(\cdot)$ denote the posterior variance corresponding to the set $\mathcal{D}_j$ obtained by sampling $NT_j$ points uniformly at random from the domain $\mathcal{X}_j$. Then, for $T_1 \geq \overline{M}(\delta)/N$ and for any $f$ satisfying Assumption 2.1, the following holds w.p. $1 - \delta$ for all epochs $j \geq 1$:*

$$\sup_{x \in \mathcal{X}_j} \sigma_j^2(x) \leq C_{f,\mathcal{X}} \cdot \frac{\gamma_{NT_j}}{NT_j}.$$

*Here $C_{f,\mathcal{X}}$ denotes a constant that depends only on $f$ and $\mathcal{X}$ and is independent of both $N$ and $T$.*

The bound on the communication cost follows directly from the following Lemmas B.3 and B.4 by noting that the communication cost in epoch $j$ satisfies $\mathcal{O}(|\mathcal{S}_j|)$.

**Lemma B.3.** *The total number of epochs in* DUETS *over a time horizon of $T$ is at most $\log(\log(\max\{N, T\})) + 4$.*

**Lemma B.4.** *Let $\mathcal{S}_j$ denote the inducing set construct in $j^{th}$ epoch, as outlined in Section 3. Then, for all epochs $j$ the following holds with probability at least $1 - \delta$,*

$$|\mathcal{S}_j| \leq C_{f,\mathcal{X}} \cdot (3 + \log(\log(\log N \log T)/\delta)) \cdot \gamma_{NT},$$

*where $C_{f,\mathcal{X}}$ is same constant as the one in Lemma B.2.*

Consider any epoch $j \geq 1$ and let $R^{(j)}$ denote the regret incurred by DUETS in this epoch. Since the agents take purely exploratory actions by uniform sampling points from the current set, we have the following crude bound $R^{(j)} \leq \Delta_j \cdot NT_j \cdot M_f$, where $\Delta_j := \sup_{x \in \mathcal{X}_j}(f(x^*) - f(x))$. The term

$NT_j \cdot M_f$ corresponds to number of points sampled during the epoch as we sample each connected component of $\mathcal{X}_j$, of which there are at most $M_f$, $NT_j$ times. For $j = 1$, we use the trivial bound,

$$\Delta_1 = \sup_{x \in \mathcal{X}} (f(x^*) - f(x)) \le 2 \sup_{x \in \mathcal{X}} f(x) \le 2B,$$

which gives us $R^{(1)} \le 2B \cdot NT_1 \cdot M_f$. On invoking Lemma B.1 for $j > 1$ we obtain,

$$R^{(j)} \le \Delta_j \cdot NT_j \cdot M_f$$
$$\le NT_j \cdot M_f \cdot \left( 8\beta(\delta') \cdot \left( \sup_{x \in \mathcal{X}_{j-1}} \sigma_{j-1}(x) \right) + \frac{4B}{T} \right),$$

where $\delta' = \dfrac{\delta}{2(\log \log NT + 4)|\mathcal{U}_T|}$. Using Lemma B.2, we can further bound this expression as

$$R^{(j)} \le \Delta_j \cdot NT_j \cdot M_f$$
$$\le NT_j \cdot M_f \cdot \left( 8\beta(\delta') \cdot C_{f,\mathcal{X}} \cdot \sqrt{\frac{\gamma_{NT_{j-1}}}{NT_{j-1}}} + \frac{4B}{T} \right)$$
$$\le M_f \cdot \left( 8C_{f,\mathcal{X}}^{1/2} \cdot \beta(\delta') \cdot \sqrt{NT\gamma_{NT_{j-1}}} + \frac{4BNT_j}{T} \right)$$
$$\le M_f \cdot \left( 8C_{f,\mathcal{X}}^{1/2} \cdot \beta(\delta') \cdot \sqrt{NT\gamma_{NT}} + \frac{4BNT_j}{T} \right).$$

In the third line, we used the inequality $\dfrac{T_j}{\sqrt{T_{j-1}}} \le \sqrt{T}$, which follows from the definition of $T_j$. In the last line, we used the fact that $\gamma_m$ is an increasing function of $m$. Thus, if $J$ denotes an upper bound on the number of epochs, we can write:

$$\sum_{j=1}^{J} R^{(j)} \le 2BM_f \cdot NT_1 + \sum_{j=2}^{J} M_f \cdot \left( 8C_{f,\mathcal{X}}^{1/2} \cdot \beta(\delta') \cdot \sqrt{NT\gamma_{NT}} + \frac{4BNT_j}{T} \right)$$
$$\le 2BM_f \cdot NT_1 + J \cdot \left( 8C'_{f,\mathcal{X}} \cdot \beta(\delta') \cdot \sqrt{NT\gamma_{NT}} \right) + \frac{4BNM_f}{T} \sum_{j=1}^{J} T_j$$
$$\le 2BM_f \cdot NT_1 + J \cdot \left( 8C'_{f,\mathcal{X}} \cdot \beta(\delta') \cdot \sqrt{NT\gamma_{NT}} \right) + 4BNM_f. \qquad (5)$$

We next optimize the length of the first epoch $T_1$ in order to achieve order optimal regret. DUETS achieves order optimal regret for $N \le \max(T, \gamma_{NT})$.

If $N < T$ we can choose $T_1 = \sqrt{\frac{T}{N}} + \overline{M}(\delta')$ where $\delta' = \frac{\delta}{2(\log \log NT + 4)}$. Left hand side of equation (5) can now be written as $\widetilde{\mathcal{O}}(\sqrt{NT\gamma_{NT}}\beta(\delta')) \equiv \widetilde{\mathcal{O}}\left( \sqrt{NT\gamma_{NT}} \left( \log \frac{T}{\delta} \right) \right)$.

If $N \le \gamma_{NT}$ we can fix $T_1 = \sqrt{T} + \overline{M}(\delta')$. We have $NT_1 \le \widetilde{O}(\sqrt{NT\gamma_{NT}})$ and the left hand-side is once again $\widetilde{\mathcal{O}}\left( \sqrt{NT\gamma_{NT}} \left( \log \frac{T}{\delta} \right) \right)$.

Note that by Lemma B.3, $J$ is upper bounded by $\log(\log N \log T) + 4$ and is thus $\widetilde{O}(1)$.

Before moving onto the proofs of Lemmas B.1 and B.3, we state two auxiliary lemmas that will be useful for our analysis.

**Definition B.5.** Let $\mathcal{D} = \{x_1, x_2, \dots, x_m\} \subset \mathcal{X}$ be a collection $m$ points and $\mathcal{S}$ be any subset of $\mathcal{D}$. Let $\sigma_{\mathcal{D}}^2(\cdot)$ denote the posterior variance corresponding to the points in $\mathcal{D}$ and $\tilde{\sigma}_{\mathcal{S}}^2(\cdot)$ denote the *approximate* posterior computed based on the points in $\mathcal{S}$. We call $\mathcal{S}$ to be an $\varepsilon$-accurate inducing set if the following relations are true for all $x \in \mathcal{X}$.

$$\frac{1-\varepsilon}{1+\varepsilon} \cdot \tilde{\sigma}_{\mathcal{S}}^2(x) \le \sigma_{\mathcal{D}}^2(x) \le \frac{1+\varepsilon}{1-\varepsilon} \cdot \tilde{\sigma}_{\mathcal{S}}^2(x).$$

**Lemma B.6** (Adapted from [5]). *Let $\mathcal{D} = \{x_1, x_2, \dots, x_m\} \subset \mathcal{X}$ be a collection $m$ points and $\mathcal{S}$ be a random subset of $\mathcal{D}$ constructed by including each point with probability $p$, independent of*

*other points. Then $\mathcal{S}$ is an $\varepsilon$-accurate inducing set with probability $1 - 4m \exp\left(-\dfrac{3p\varepsilon^2}{8\sigma_{\max}^2}\right)$, where* $\sigma_{\max}^2 = \sup_{x \in \mathcal{X}} \sigma_{\mathcal{D}}^2(x)$.

**Lemma B.7.** *Let* DUETS *be run with a choice of $p_0 = 72 \log(4NT/\delta')$. Then, for all epochs $j \geq 1$, the global inducing set $\mathcal{S}_j$ is $0.5$-accurate with probability $1 - \delta$.*

*Proof.* The statement is an immediate consequence of Lemma B.6 with the given choice of parameter $p_0$. □

We are now ready to prove Lemmas B.1 and B.3.

## B.1 Proof of Lemma B.1

To ensure that the bound holds over the entire arm set we adopt a standard discretization assumption:

**Assumption B.8.** *For each $r \in \mathbb{N}$, there exists a discretization $\mathcal{U}_r$ of $\mathcal{X}$ with $|\mathcal{U}_r| = \mathrm{poly}(r)^4$ such that, for any $f \in \mathcal{H}_k$, we have $|f(x) - f([x]_{\mathcal{U}_r})| \leq \dfrac{\|f\|_{\mathcal{H}_k}}{r}$, where $[x]_{\mathcal{U}_r} = \arg\min_{x' \in \mathcal{U}_r} \|x - x'\|_2$.*

The existence of the discretization $\mathcal{U}_r$ in Assumption B.8 has been justified and adopted in previous studies [28, 30]. In particular, the popular class of kernels like Squared Exponential and Matérn kernels are known to be Lipschitz continuous, in which case a $\varepsilon$-cover of the domain with $\varepsilon = \mathcal{O}(1/r)$ is sufficient to show the existence of such a discretization

Consider any epoch $j \geq 2$ and let $x \in \mathcal{X}_j$. Let $\Delta(x) := f(x^*) - f(x)$. We will obtain a bound on $\Delta(x)$ for any general $x$ in order establish the bound on $\Delta_j$. Using the discretization from Assumption B.8 for $\mathcal{X}_j$, we obtain,

$$
\begin{aligned}
\Delta(x) &= f(x^*) - f(x) \\
&\leq f(x^*) - f([x^*]_{\mathcal{U}_T}) + f([x^*]_{\mathcal{U}_T}) - (f(x) - f([x]_{\mathcal{U}_T})) - f([x]_{\mathcal{U}_T}) \\
&\leq f([x^*]_{\mathcal{U}_T}) - f([x]_{\mathcal{U}_T}) + \frac{2B}{T}.
\end{aligned}
$$

Using the result from [21], we obtain the following high probability bound that holds with probability $1 - \delta$:

$$
\begin{aligned}
\Delta(x) &\leq f([x^*]_{\mathcal{U}_T}) - f([x]_{\mathcal{U}_T}) + \frac{2B}{T} \\
&\leq \tilde{\mu}_j([x^*]_{\mathcal{U}_T}) + \beta(\delta')\tilde{\sigma}_j([x^*]_{\mathcal{U}_T}) - \tilde{\mu}_j([x]_{\mathcal{U}_T}) + \beta(\delta')\tilde{\sigma}_j([x]_{\mathcal{U}_T}) + \frac{2B}{T} \\
&\leq \tilde{\mu}_j(x^*) - \tilde{\mu}_j(x) + \beta(\delta')\tilde{\sigma}_j([x^*]_{\mathcal{U}_T}) + \beta(\delta')\tilde{\sigma}_j([x]_{\mathcal{U}_T}) + \frac{4B}{T},
\end{aligned}
$$

where we again used Assumption B.8 in the last step. We claim that $x^* \in \mathcal{X}_{j-1}$ for all $j \geq 2$. Assuming this claim this true, we can bound the above expression as

$$
\begin{aligned}
\Delta(x) &\leq \sup_{x \in \mathcal{X}_{j-1}} \tilde{\mu}_j(x') - \tilde{\mu}_j(x) + \beta(\delta')\tilde{\sigma}_j([x^*]_{\mathcal{U}_T}) + \beta(\delta')\tilde{\sigma}_j([x]_{\mathcal{U}_T}) + \frac{4B}{T} \\
&\leq 2\beta(\delta')\sigma_{j,\max} + \beta(\delta')\tilde{\sigma}_j([x^*]_{\mathcal{U}_T}) + \beta(\delta')\tilde{\sigma}_j([x]_{\mathcal{U}_T}) + \frac{4B}{T},
\end{aligned}
$$

where we used the update condition (Eqn. (3)) in the second step. Since $\mathcal{S}_j$ is $0.5$-accurate (Lemma B.7), we have $\tilde{\sigma}_j^2(x) \leq 3\sigma_j^2(x) \leq 3\sigma_{j,\max}^2$. On plugging this back into the above equation, we obtain,

$$
\Delta(x) \leq 8\beta(\delta')\sigma_{j,\max} + \frac{4B}{T}.
$$

The statement of the lemma follows by $\Delta_j = \sup_{x \in \mathcal{X}_j} \Delta(x)$.

---

[4]The notation $g(x) = \mathrm{poly}(x)$ is equivalent to $g(x) = \mathcal{O}(x^k)$ for some $k \in \mathbb{N}$.

We prove our claim $x^* \in \mathcal{X}_j$ for all $j \geq 1$ using induction. Clearly, $x^* \in \mathcal{X}_1 = \mathcal{X}$, by definition. Assume $x^* \in \mathcal{X}_{j-1}$ for some $j \geq 2$. Fix an arbitrary $x \in \mathcal{X}_{j-1}$, from the confidence bound lemma we have:

$$\mu_{j-1}(x) - \mu_{j-1}(x^*) \leq (f(x) - f(x^*)) + \beta(\delta')(\sigma_j(x) + \sigma_j(x^*)) \leq 2\sigma_{j-1.\max}(x),$$

where the second inequality follows as $f(x) \leq f(x^*)$. As the inequality holds $\forall x \in \mathcal{X}_{j-1}$ we must have:

$$\sup_{x \in \mathcal{X}_{j-1}} \mu_{j-1}(x) - \mu_{j-1}(x^*) \leq 2\sigma_{j-1.\max}(x)$$

and thus indeed $x \in \mathcal{X}_j$.

## B.2  Proof of Lemma B.3

We define $E(s) := \min\{j : T_j \geq T/4 \mid T_1 = s\}$. Note that $T_j$ is an increasing function of $j$. Since $T_{E(s)} \geq T/4$, we can conclude that $E(s) + 4$ is an upper bound on the number of epochs. Thus, we focus on bounding $E(s)$. We first show that $E(s)$ is a non-decreasing function of $s$.

To that effect, we claim that for $j \geq 2$ the epoch lengths satisfy the relation $T_j \geq T^{1-2^{-j+1}} \cdot T_1^{2^{-j+1}}$. This relation follows immediately using induction. For the base case, note that $T_2 \geq T^{1/2} \cdot T_1^{1/2}$, by definition. Assume that the relation holds for $j - 1$. Thus,

$$T_j \geq T^{1/2} \cdot T_{j-1}^{1/2} \geq T^{1/2} \cdot T^{1-2^{-(j-1)+1-1}} \cdot T_1^{2^{-(j-1)+1-1}} \geq T^{1-2^{-j+1}} \cdot T_1^{2^{-j+1}}. \tag{6}$$

Since $T_j$'s are lower bounded by an increasing function of $T_1$, the number of epochs $E(s)$ is a non-increasing function of $s$. Since $T_1 \geq \frac{T}{N}$, $E\left(\frac{T}{N}\right)$ is an upper bound on the number of epochs for all choices of $T_1$.

Let $j^* = \max\{\log(\log(T)), \log(\log(N))\}$. Using the above relation for $T_j$ from Eqn. (6) and the lower bound on $T_1$, we have,

$$T_{j^*} \geq T \cdot N^{-2^{1-j}} = T \cdot \left(2^{-\frac{\log N}{2^j}}\right)^2 \geq T \cdot 2^{-2}$$

We can hence conclude that $T_{j^*} \geq T/4$, which implies that $E(T_1) \leq j^*$ for all permissible choices of $T_1$. Consequently, the number of epochs are bounded as $\log(\log(\max\{N, T\})) + 4$.

## B.3  Proof of Lemma B.4

For all epochs $j \geq 1$, recall that the inducing set is constructed by including each point from $\mathcal{D}_j$ with probability $p_j$, independent of other points. Thus, $|\mathcal{S}_j|$ is a binomial random variable with parameters $|\mathcal{D}_j| = NT_j$ and $p_j$. Using the Chernoff bound for Binomial random variables, we can conclude that

$$\Pr(|\mathcal{S}_j| > (1+\varepsilon)NT_j p_j) \leq \exp\left(-\frac{\varepsilon^2 NT_j p_j}{2+\varepsilon}\right).$$

Invoking the bound with $\varepsilon = 2 + \log(1/\delta')$, with $\delta' = \delta/(\log\log(NT) + 4)$ yields that the following relation holds with probability $1 - \delta'$:

$$\begin{aligned}
|\mathcal{S}_j| &\leq (3 + \log(1/\delta')) \cdot NT_j p_j \\
&\leq (3 + \log(1/\delta')) \cdot NT_j \cdot p_0 \sigma_{j,\max}^2 \\
&\leq (3 + \log(1/\delta')) \cdot NT_j p_0 \cdot C_{f,\mathcal{X}} \cdot \frac{\gamma_{NT_j}}{NT_j} \\
&\leq (3 + \log(1/\delta')) p_0 \gamma_{NT},
\end{aligned}$$

where we used Lemma B.2 in the third step and monotonicity of $\gamma_m$ in the last step. On taking a union bound over all epochs and using the bound on the number of epochs from Lemma B.3, we conclude that for all epochs $j$, $|\mathcal{S}_j| = \tilde{\mathcal{O}}(\gamma_{NT})$ with probability $1 - \delta$.

# C  Shared Randomness in DUETS

The setup in DUETS where the server and each agent have access to shared randomness is similar to the class of *public coin protocols* that have been extensively studied in field of Information Theory. Such class of algorithms assume that agents and the server share a common source of randomness that is independent of the environment and comes at no communication cost between agents and the server Park et al.[19], Jayadev et al.[2], Jayadev et al.[1], Wang et al.[31]. Public coin protocols can concretely be implemented as an agent-server agreement on the generating seeds before the start of the algorithm, which takes at most $\mathcal{O}(1)$ communication cost.

We stress that the source of shared randomness between the agent and the server is established before any interaction with the environment and thus cannot carry any information relevant to learning the reward function. It simply allows agents to follow an agreed on non-adaptive randomized strategy while querying the environment similar to the randomized strategy outlined in Jayadev et al. [2] for choice of communication channels.

# D  Empirical Studies

In this section, we provide additional details about our empirical studies along with results on benchmark functions.

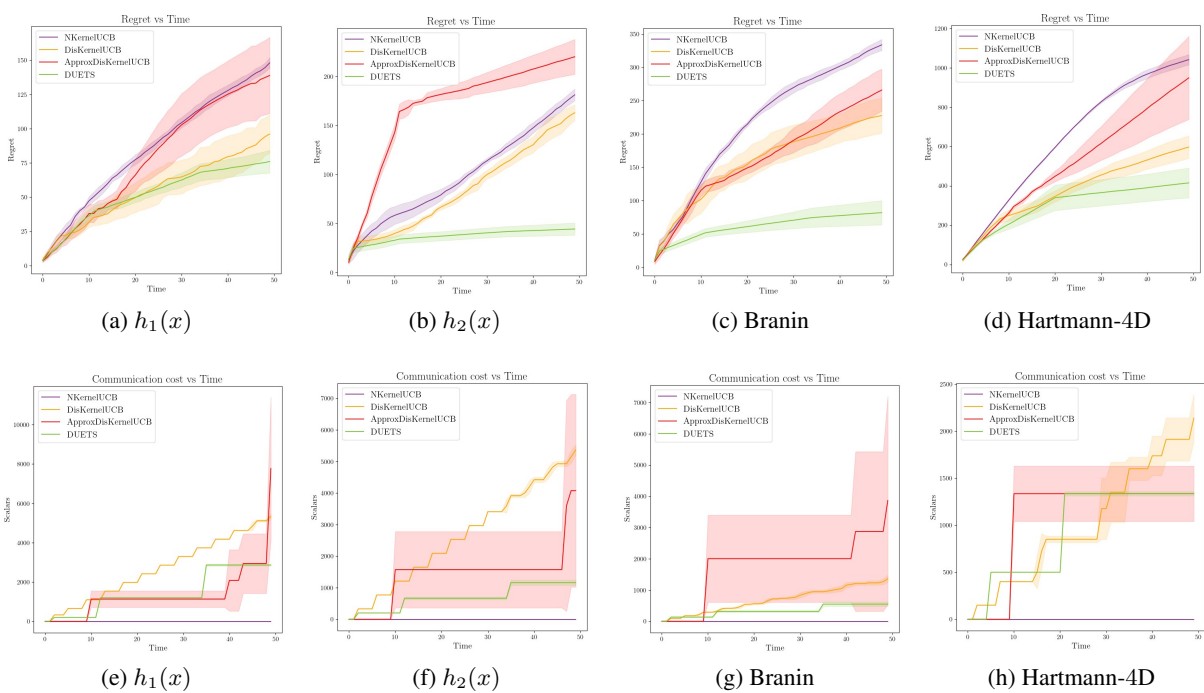

Figure 1: Cumulative regret (Fig. (1a-1d) and communication cost (1e-1h) for all algorithms across different benchmark functions averaged over 5 Monte Carlo runs. The shaded region represents error bars corresponding to one standard deviation. DUETS obtains a superior performance, both in terms of regret and communication cost, over other algorithm across all functions.

We perform several empirical studies to corroborate our theoretical findings. We compare the regret performance and communication cost of our proposed algorithm, DUETS, against three baseline algorithms — DisKernelUCB, ApproxDisKernelUCB and N-KernelUCB. The first two are distributed kernel bandits algorithms proposed in Li et al. [15]. N-KernelUCB is a baseline algorithm considered in Li et al. [15] where each agent locally runs the GP-UCB algorithm  Gopalan and Chowdhury [7] with no communication among the agents.

We compare the performance of all the four algorithm across four benchmark functions. The first two are synthetic functions $h_1, h_2 : \mathcal{B} \to \mathbb{R}$ considered in Li et al. [15], where $\mathcal{B}$ denotes the unit ball centered at origin in $\mathbb{R}^{10}$. The functions are given by:

$$h_1(x) := \cos(3x^\top \theta^\star); \quad h_2(x) := (x^\top \theta^\star)^3 - 3(x^\top \theta^\star)^2 + 3(x^\top \theta^\star) + 3.$$

For both the functions $\theta^\star$ is randomly chosen from the surface of the unit ball $\mathcal{B}$. The other two functions are Branin [4, 20] and Hartmann-4D [20], which are commonly used benchmark functions for Bayesian Optimization. The Branin function is defined over $\mathcal{X} = [0, 1]^2$ while the Hartmann-4D function is defined over $\mathcal{X} = [0, 1]^4$.

We consider a distributed kernel bandit described in Section 2 with $N = 10$ agents. For all the experiments, we use the Squared Exponential kernel. The length scale was set to 0.2 for Branin and to 1 for all other functions. The observations were corrupted with zero mean Gaussian noise with a standard deviation of 0.2. The parameter $D$ for ApproxDisKernelUCB and DisKernelUCB was set to 20 and 10 respectively. For DUETS , we set $T_1 = 2$ and $p_0 = 10$. The parameter $\beta$ was selected using a grid search over $\{0.2, 0.5, 1, 2, 5\}$ for all the algorithms. All the algorithms were run for $T = 50$ time steps. We averaged the cumulative regret and the communication cost incurred by different algorithms over 5 Monte Carlo runs.

The cumulative regret incurred by different algorithms across the different benchmark function are shown in the top row of Figure 1. The bottom row consists of the corresponding plots for the communication cost incurred by the different algorithm. The shaded regions denotes error bars upto standard deviation on either side of the mean value. As evident from the plots, DUETS achieves a significantly lower regret as compared to all other algorithms consistently across benchmark functions. DUETS also incurs a smaller communication overhead as compared to other algorithms, corroborating our theoretical results.

All simulations were run on an Intel(R) Xeon(R) E-2176M CPU@ 2.70GHz with 6 cores with no GPU's. Runtime of 5 Monte-Carlo-simulations for all 4 algorithms on a single data-set took roughly 2 hours of CPU time. Estimated runtime of all conducted experiments is 8 hours.

