# OpenReview forum: "Order-Optimal Regret in Distributed Kernel Bandits using Uniform Sampling with Shared Randomness"
_NeurIPS.cc/2024/Workshop/BDU — NeurIPS BDU Workshop 2024 Poster_

### Official Review · Reviewer_RL76 · 2024-09-16
**This paper presents a novel algorithm, DUETS, for distributed kernel bandit problems. It tackles the crucial trade-off between learning speed and communication efficiency, achieving optimal regret with sublinear communication costs. The key innovations are the use of uniform sampling for exploration and shared randomness for efficient information exchange. The paper provides rigorous analysis and demonstrates empirical performance on benchmark functions.**

**Rating:** 8
**Confidence:** 4

**Review:**

Pros:

Novel Approach: The paper presents a novel algorithm, DUETS, for distributed kernel bandits, which addresses the critical challenge of balancing learning speed and communication efficiency.

Strong Theoretical Results: The authors provide rigorous theoretical analysis, showing that DUETS achieves the optimal regret order with a sublinear communication cost. These are demonstrating the algorithm's efficiency.

Empirically Supported: The empirical studies provide strong evidence supporting the algorithm's performance. They compare DUETS with existing approaches and show its superiority in terms of regret and communication cost.

Cons:

The assumption about the smoothness and structure of the reward function (Assumption 2.1) is quite restrictive. It's unclear how well the algorithm would perform in scenarios where this assumption doesn't hold.

Limited Scope: The focus on fixed-arm settings might limit the generalizability of the approach to more complex scenarios with contextual bandits or dynamic arm sets.

---

### Official Review · Reviewer_6uKF · 2024-09-26
**This paper proposed a new algorithm DUETS with better regret bound and communication cost bound on distributed kernel bandit problem. But it does not compare the recent progress and the achievement is slightly unclear in the simulation.**

**Rating:** 4
**Confidence:** 3

**Review:**

This paper proposed a new algorithm DUETS (Distributed Uniform Exploration of Trimmed Sets) to tackle the following challenges:
1. potentially infinite-dimensional communication cost on distributed kernel bandit problem
2. communication cost grows linearly in both N (number of agents) and T (time spent) on existing solutions
3. how to achieve the performance of a centralized learner with NT query points

Pros:
1. The new algorithm has better regret bound and communication cost bound with solid theoretical derivations.
2. The novel algorithm shows better regret and communication cost during the simulation based experiments.

Cons:
1. The experiment does not include a single bandit learner benchmark. Therefore it is not very clear to understand the quantitative achievements of the proposed algorithm.
2. The literature review is not complete. The paper does not include the recent progress which improves the regret bound and communication cost bound against the same benchmarks (example is attached at the end).
3. The paper discusses the communication cost in
    1) a synchronized flavor and
    2) without any communication modeling (for example, communication lost).
  In the real world this setting is too optimistic. Typically this is not a problem as a novel algorithm. However as a-synchronized algorithms are developed recently, the synchronized algorithm with same assumption is less practical and hence less flavored.

example of recent progress on same topic: Learning Kernelized Contextual Bandits in a Distributed and Asynchronous Environment

---

### Decision · Program_Chairs · 2024-10-09

Accept (Poster)